# Use of Ashes from Lignite Combustion as Fillers in Rubber Mixtures to Reduce VOC Emissions

**DOI:** 10.3390/ma14174986

**Published:** 2021-08-31

**Authors:** Miroslawa Prochon, Dariusz Bieliński, Paulina Stepaniak, Magdalena Makowicz, Dominik Pietrzak, Oleksandra Dzeikala

**Affiliations:** Institute of Polymer and Dye Technology, Faculty of Chemistry, Lodz University of Technology, Stefanowskiego 12/16, 90-924 Lodz, Poland; dariusz.bielinski@p.lodz.pl (D.B.); 200249@edu.p.lodz.pl (P.S.); 198881@edu.p.lodz.pl (M.M.); dominik.pietrzak@p.lodz.pl (D.P.); oleksandra.dzeikala@p.lodz.pl (O.D.)

**Keywords:** ash after lignite combustion, filler, rubber mixtures, volatile organic compounds (VOC)

## Abstract

This paper presents the use of ashes from brown coal combustion (BCA) as fillers in rubber mixtures, to reduce the emission of volatile organic compounds. Two types of ash, BCA1 and BCA2, were selected as fillers for styrene–butadiene rubber (SBR). The ashes were produced during the treatment of brown coal at the Bełchatów Power Plant in the years 2017 and 2018. The morphology and chemical composition of the ash were tested. Morphology studies using scanning microscopy showed differences in the grain sizes of the ashes, and EDS analysis showed a difference in their chemical compositions. Vulcanizates with different weight proportions of the individual ashes were produced. Mixtures were made with the addition of 10–30 pts. wt. ashes per 100 g of SBR. The addition of BCA1 ash at 10 and 30 pts. wt. reduced the emission of volatile organic compounds (VOC) while maintaining the good strength properties of the mixtures.

## 1. Introduction

During the process of coal combustion, significant amounts of by-products are formed, including fly ash, furnace slag, and harmful chemical compounds in the form of gases (including CO_2_, NO_x_, and sulfur compounds) [1,2,3,4,5,6,7]. A large proportion of this furnace waste (ash and slag) is stored. The storage of ashes is not ideal due to the significant risk of atmospheric factors leading to the release of dust particles into the air. Some uses for ashes have been found. An interesting application of ashes is in the production of electrodes in lithium-ion batteries [8]. Energy waste is also used in mining (as flooring components, to strengthen the rock mass) [9], agriculture (for the production of fertilizers, soil deacidification), civil engineering (for soil stabilization, construction of embankments, flood sealing), and in environmental protection, where energy waste is used to neutralize sewage and purify exhaust gases [10].

Ashes can also be used in construction, as additives in cement and concrete [11,12,13,14]. However, not all ashes can be exploited in this way, because their chemical composition is not in line with the relevant standards. For example, A. Michalik et al. [15] developed a method of improving the properties of fly ash as an additive for cements. However, some of the ash remains and is treated as waste. Limestone ash, after the separation of the carbon fractions and silica ash, is also used in the construction industry as a concrete mass.

The possibilities of using all of an ash are quite limited, due to the problematic phase and chemical compositions of ashes, as well as their physicochemical properties. Finding a way to recycle ash could significantly reduce the amount of landfilled waste. The scale of the problem can be seen in the case of the Bełchatów Power Plant (Poland), which is the largest thermal power plant in Europe. This power plant alone produces and stores approximately three million tons of ash and slag annually [14]. The ashes produced as a result of various methods of coal combustion show significant differences in terms of their phase and chemical compositions, degrees of dispersion, and particle morphologies. The use of ashes in this form is difficult, requiring research to determine their specific properties and identify suitable applications [6].

An innovative idea may be to use ashes from brown coal combustion as fillers in elastomeric blends. Plastics are widely used in many industries, including in the automotive industry as car cabin accessories, everyday objects, interior fittings, etc., [16]. However, plastics are a potential source of volatile organic compounds (VOCs), which are released in the processes of degassing materials [17]. Volatile organic compounds are known to adversely affect human health; they constitute about 73% of carcinogenic compounds on the list of toxic compounds [18]. The highest concentrations of VOCs are recorded indoors. It is estimated that adults spend about 80% of their time indoors, which puts them in prolonged contact with these toxic compounds. Therefore, manufacturers should strive to minimize VOC emissions both during the production process and from plastic products.

Styrene–butadiene rubber (SBR) is a type of polymeric compound based on rubber that is commonly used in the tire industry [19]. SBR rubber is also used as a waste material for the growth of carbon nanomaterials, such as nanofibers or nanotubes. Small granules of rubber are subjected to pyrolysis at a temperature of 1000 °C. The pyrolysates are mixed with gases containing burned SBR and oxygen. Combustion products are used to synthesize carbon nanomaterials (CNMs) in the presence of catalysts. CNMs have a characteristic structure of approximately 30–100 microns in diameter and 10 microns in length. Due to their unique mechanical, electrical, and thermal properties, CNMs have a number of potential added value applications [20].

In a study by X. Ren and E. Sancaktar [21], by-products from the energy industry in the form of ashes were used as a fly reinforcement, replacing typical fillers such as carbon black and silica in rubbers. The partial addition of up to 10 pts. wt. of fly ash in combination of 54 and 4 pts. wt. of carbon black and silica fillers, respectively, resulted in increased elongation at break, better adhesion to the steel reinforcement cord, and improved wet grip. Lower rolling resistance was also observed, which was attributed to the more effective reinforcing effect of the silica contained in the fly ash.

Here, we present a new method of using ashes as fillers for rubber mixtures. The ashes were produced during combustion of lignite at the Bełchatów Power Plant in 2017 (BCA1] and 2018 (BCA2]. An additional aim was to reduce the emission of VOCs from the mixtures produced during the vulcanization process.

## 2. Materials and Methods

### 2.1. Materials

#### 2.1.1. Rubber and Other Ingredients

The rubber mixtures were made using SBR butadiene–styrene rubber (KER 1500, Synthos S.A. Oświęcim, Poland) as the matrix for each mixture. The cross-linking unit included sulfur, as the cross-linking substance (S_8_ orthorhombic; density 2.07 g/cm^3^, Siarkopol Tarnobrzeg Sp. z o.o., Tarnobrzeg, Poland), zinc oxide as the activator (zinc white, ZnO, Huta Będzin, Poland), and CBS (Bestgum Polska Sp.z o.o., Rogowiec, Poland) as the accelerator.

#### 2.1.2. Ash after Burning Brown Coal (BCA)

Technical carbon black N330 (75 × 10^3^ m^2^/kg, Fermintrade, Konin, Poland) and ash from Elektrownia Bełchatów (PGE Górnictwo i Energetyka Konwencjonalna SA, Elektrownia Bełchatów, Poland) produced in 2017 (BCA1) and 2018 (BCA2) were used as the fillers. The mixtures also included technical stearin (Torimex-Chemicals Ltd. Sp.z o.o., Konstantynów Łódzki, Poland).

### 2.2. Preparation of Composites

After the preparation step (see Section 2.1.2), the BCA filler was introduced at different pts. wt. into the SBR. The rubber mixtures were prepared using a mixing mill (Bridge mill, London, UK) with a roller temperature of 27–37 °C and a friction/friction of 1.1. The rolling mill parameters were as follows: roll length L = 450 mm; roll diameter D = 200 mm; rotational speed of the front roller Vp = 20 (rpm); width of the gap between rollers 1.5–3 mm. The mixtures were prepared for 6 min and then wrapped in foil and stored at 2–6 °C. Table 1 compares the tested rubber mixtures. The mixtures were made with carbon black N330 or a mixture of carbon black with 10, 20, and 30 pts. wt. ashes. Tests were carried out at room temperature under normal pressure conditions.

## 3. Research Techniques

### 3.1. Researching Ashes

To determine the structure of the ashes, microscopic pictures were taken using a Keyence VHX 1000 optical microscope (Keyence International, Mechelen, Belgium). To determine the share of particles of a given size in the tested ashes, a sieve analysis was performed. For this purpose, an AS200Control device (Retsch GMBH, Haan, Germany) with EasySieve software (Retsch GMBH, Haan, Germany) was used, consisting of eight sieves with decreasing mesh sizes: 4.00 mm; 2.00 mm; 1.00 mm; 0.50 mm; 0.25 mm; 0.125 mm; 0.063 mm; 0.045 mm. The measurements lasted 3 min, during which the computer program processed the results from the testing device.

### 3.2. Making Mixtures

The ingredients were mixed together in the following order: rubber → ZnO → stearin → soot → CBS → sulfur. A rolling mill (Bridge type milling machine, London, UK) was used with a roll length of 150–300 mm, temperature 30 °C, rotational speed 15–13 rpm. The cross-linking kinetics of the SBR compounds were tested using an ALPHA MDR 200 rheometer with a moving nozzle (MonTech MDR 300, Buchen, Germany) at a temperature of 160 °C for 30 min, in accordance with PN-ISO 3417:1994. Hardening (τ90) was performed in a standard electrically heated hydraulic press (PH-2PW90, ZUP Nysa Sp. z o.o. Nysa, Poska) at a temperature of 160 °C, pressure 32 MPa, dimensions of heating plates 400 mm × 400 mm. After vulcanization, the samples were prepared for strength tests in accordance with the PN-EN ISO 3167:2014-09 standard. Paddle-shaped samples were cut out using a ZCP020 press (Zwick/Roell GmbH & Co. KG., Ulm, Germany).

The mechanical properties of the prepared composites were tested on a universal testing machine Zwick model 1435 (Zwick/Roell, Radeberg, Germany), in accordance with the PN-ISO 37:2007 standard. Paddle-shaped samples were subjected to stretching at room temperature at a constant speed of 500 mm/min and with a preforce of 0.3 N. The tensile strength (TSb) and the percentage elongation at break (E_b_) were determined. A hardness test (H, °Sh) was performed using an electronic Shore A hardness tester (Zwick/Roell, Herefordshire, UK) with a pressing force of 12.5 Nz, in accordance with PN-80C-04238 [22].

The polymer–solvent interaction parameter was determined by the equilibrium swelling method according to PN ISO 1817:2001/ap1:2002 [23] (0.378 for SBR rubber in a toluene solvent). Samples were then allowed to evaporate in air for 72 h and reweighed.

Cross-link density was calculated as the volume fraction of rubber in the swollen material, and vs. = 106.3 mol/cm^3^ for the molar volume of the solvent (toluene) solvent [24,25]. 

Cross-link density (*υ*), was calculated on the basis of the Flory–Rehner equation:(1)$$v=\frac{\ln\left(1-V_{r}\right)+V_{r}+\mu V_{r}^{3}}{V_{0}\left(V_{r}^{\frac{1}{3}}-\frac{V_{r}}{2}\right)}$$
where *μ* is the Huggins parameter for the uncrosslinked polymer–solvent system and *Vr* is the molar volume of the swelling solvent.

Rheological tests were performed using a rotational rheometer (ARES-G2 TA Instruments, New Castle, DE, USA). Stress measurements were performed regardless of the applied shear deformation, with a torque range from 0.05 µNm to 200 mNm, force range of 0.001–20 N, temperature range from −150 °C to +600 °C, cooling with liquid nitrogen, DETA module for dielectric measurements.

To test the effect of elevated temperature, the samples were placed in a thermal chamber (Binder GmbH, Tuttlingen, Germany) at 70 °C for 7 days. Three mixtures were selected and subjected to 30-day biological aging in soil. The soil samples were placed in a MEMMERT climate chamber (HPP 108 Memmert GmbH, Schwabach, Germany) for 14 days at 30 °C and 80% air humidity. The aged samples were then subjected to strength tests. The results were compared with those for samples that had not been aged. Examination of the surface morphology of the rubber mixtures was carried out using a Zeiss Ultra Plus scanning electron microscope (ZEISS, Carl Zeiss AG, Oberkochen Germany). A Nicolet 6700 FTIR spectroscope (Thermo Scientific, Waltham, MA, USA) with Fourier transform was used to analyze the chemical bonds present in the ashes and the obtained vulcanizates, using the total internal reflection method with the attenuated total reflection (ATR) attachment. Measurements were carried out in the range of 400–4000 cm^−1^ for the ashes and three selected vulcanizates.

The VOCs were studied using ion mobility spectroscopy with gas chromatography, during vulcanization in a tightly closed reactor. The heating time of the mixture at 160 °C was 2 × t90 (two times optimum cure time) in order to be sure that the vulcanization occured. The collected gases were diluted and sent to an MCC-IMS instrument (G.A.S. Gesellschaft für Analytische Sensorsysteme GmbH, Dortmund, Germany). Tests were carried out for a mixture without the addition of ash and for two mixtures with CBS1 ash (addition of 10 or 30 pts. wt. ash). Thermal analysis by differential scanning calorimetry (DSC) was carried out in order to determine the glass transition temperature (Tg) of the selected vulcanizates. The tests were carried out using a DSC_1_ analyzer (Mettler Toledo, Netzsch, Switzerland) calibrated with standards (indium, zinc) at a heating rate of 10 °C/min. The SBR samples were heated from −150 °C to 350 °C under nitrogen atmosphere.

## 4. Results and Discussion

### 4.1. Characterization of BCA

It is known that the shape, particle size, and specific surface of a filler have a decisive influence on the strength of rubber–filler connections. Ash morphology was assessed on the basis of photographs taken with an optical microscope, shown in Figure 1. The ash agglomerates are visible as irregularly structured spherical particles, with a wide size distribution from several hundred nanometers to several micrometers. The grains in the BCA2 ash are much finer. The older ash contained larger particles with greater size variation.

Table 2 shows that the largest share of BCA1 ash (74.7%) was between 63–250 µm, whereas BCA2 ash contained mostly fractions below 45–250 µm (76.9%).

### 4.2. Characterization of SBR/BCA Composites

#### 4.2.1. Vulcanization Kinetics

Rheometric measurements were used to determine the vulcanization time and the increase in the torsional moment ΔM. The results are presented in Table 3. The BCA fillers clearly influenced the cross-linking properties of the SBR blends. The kinetic parameters of the BCA-containing compounds were different from those of the blank compound. The addition of ash to the SBR compound resulted in lower viscosity and stiffness, which is reflected in the torque value (ML). The increase in torque decreases as the weight fraction of ash increases, leading to a visible deterioration of the mechanical parameters of vulcanizates. The unfilled sample showed a longer cure time than the filled composites.

The vulcanization times increased with increasing BCA1 concentrations, reaching the highest value at 30 phr (τ_90_ = 12.99). Subsequent studies clearly showed that the ash belongs to the group of inactive fillers, with lower increases of turbulent moments for the composition with the same content compared to the noninvasive composite. As a consequence, it leads to a visible deterioration of the mechanical parameters of vulcanizates.

#### 4.2.2. Rheological Analysis

Rheological analysis is applicable to all materials with visco-elastic properties and describes the relationship between forces and deformation as a function of time. Figure 2 shows the dependence of changes in stress viscosity on the shear rate for individual SBR elastomer composites filled with ash. The dynamic viscosity values changed as a result of introducing ash fillers into the SBR rubber matrix. Irrespective of the type of ash used, the shear modulus decreased as the proportion of the introduced filler was increased from 10 to 20 pts. wt. This correlates with the rheometric properties (Section 4.2.1).

The use of the ash as a filler changed the viscosity of the composites. The dynamic viscosity of the mixtures was almost half that of the pure SBR sample, which may suggest the formation of connections between the mineral filler and the SBR matrix, creating a secondary structure containing mineral particles associated with the remaining components of the mixtures. As reported in [26], this may be caused by changes in deformation caused by the material microstructure, i.e., increased susceptibility to cracking and thus reduced hardness, as well as the formation of weak physical interactions connecting adjacent filler clusters.

#### 4.2.3. Cross-Linking Density

The cross-linking densities of the prepared mixtures are presented in Table 4. The reference sample had the highest cross-linking density. The samples doped with BCA1 ash did not show any particular tendency, while the cross-link densities of the polymers filled with BCA2 ash increased with higher ash contents. The cross-linked density of the SBR1 (20) and SBR2 (10) samples was 53% lower than the cross-link density of the reference sample.

#### 4.2.4. FTIR

The composites were subjected to FTIR analysis to characterize the bonds present in the ash samples and in the vulcanizates. Figure 3 shows a comparison of BCA1 and BCA2 ashes, while Figure 4 and Figure 5 show examples of spectra for SBR, SBR1 (10), and SBR1 (30) containing different amounts of ash.

The spectra presented in Figure 3 show valence vibrations of the bands coming from the side chains and hydroxyl (–OH) groups in the region from 3200 to 3500 cm^−1^. At 1706 cm^−1^, a vibration band appears (C=O). The effect of silica on interactions with other components of the BCA powder, and thus on other interactions, was manifested by shifting of the absorption bands, reducing their intensity, etc. The characteristic absorption band at 1140 cm^−1^ can be attributed to the possible interaction of Si with the protein-like system –Si–O. There is also a characteristic broad absorption band for BCA2 C-O powder at about 843 cm^−1^. The bands between 780 and 650 cm^–1^ indicate the presence of Si–O–Si bonds. A small band was observed in the range from 3600 to 3550 cm^−1^, which is related to the vibration of the –OH group in the BCA1 sample. The Al-O compound appears in the range of 920–910 cm^−1^ for both types of ashes.

As shown in Figure 4, at 2885.07 cm^−1^ visible vibrations of the C–H groups are present in the structure of the SBR aromatic styrene ring [27]. At 2100 cm^−1^ there is a C=C bond derived from rubber butadiene. The intensity of the BCA bands changes with the amount of ash, although these changes are not as significant. Small changes are visible in the dactyloscopic area related to a slight increase in the intensity of the bands in the range of 1500–1400 cm^−1^ and 500 cm^−1^. The first are characterized by stretching vibrations of the –C=C– aliphatic grouping and bending vibrations = C–H. However, the second is also the confirmation of the occurrence of stretching vibrations of the aromatic ring v (C=C), which come from styrene. Compared to the spectrum of BCA alone (Figure 4 and Figure 5), the intense band at 773 cm^−1^ derived from silicates significantly reduces in intensity, which may indicate possible interaction with Si–O–C groups in the SBR aromatic rubber ring.

The band at 953 cm^−1^ is related to the absorption of stretching vibrations from the C=C groups in the aromatic ring, which become much more intense as the amount of BCA increases. This may indicate some interaction between the filler and the elastomer matrix. The prominent peak at about 901 cm^−1^ indicates the presence of Al–O groups.

In the spectra of the mixtures after temperature aging (Figure 5), a change in the intensity of certain peaks can be observed. In the case of the signal from the C=C groups, as the share of ash in the mixture increased a decrease in the band intensity was observed at the wavenumber of 2113 cm^−1^ and around 974 cm^−1^. This was probably related to the breaking of double bonds during aging.

#### 4.2.5. Mechanical and Hardness Tests

The influence of BCA on the mechanical properties of the SBR vulcanizates is presented in Table 5. A hardness test was performed to determine the effect of the amount of added filler on the hardness of the rubber mixtures. The samples with the addition of ash showed a decrease in hardness in the range of 8.7–14.8%. The smallest decrease was observed for the SBR2 sample (20), and the largest decrease for the SBR2 sample (30). There was no downward trend in hardness as a function of sample composition (addition of ash). This parameter is related to the cross-linking density of composites and confirms the results of the swelling measurements and rheometric tests.

The results of the tensile strength tests show a decrease in breaking strength as a result of doping the rubber mixtures with ash. The SBR control sample, without the addition of ash fillers, had the highest tensile strength. The addition of ashes caused a decrease in the tensile strength of the mixtures. The smallest decrease was observed for the SBR1 sample (10). Elongation at break increased by between 12.6% and 30% following the addition of ash fillers. The highest elongation was achieved by the SBR1 sample (30), with an increase of 30% compared to the control (without filler). The only decrease in elongation in comparison to the SBR control sample occurred in the case of SBR1 (10). The values for this sample decreased with further additions of filler. This may be related to the lack of the formation of permanent molecular interactions in the form of permanent bonds, for example covalent or ionic bonds, although FTIR analysis indicated there may be weak range metallic intermolecular interactions.

A strength test was carried out on the samples after thermo-oxidative aging. In the case of thermo-oxidative treatment, the general trend remained unchanged. The highest stresses were recorded for the SBR samples, and the lowest stresses for SBR1 (30). The decreases in the elongation at break values were comparable for each of the samples. The stress at break value for the reference sample decreased by 15% compared to the sample not subjected to thermo-oxidative aging. The average reduction in tensile strength for samples filled with ash was 23%. The elongation at break value for the reference sample decreased by 41% after temperature aging. The mean decrease in elongation at break values for the samples filled with ash was 33%. The lifetime of elastomers, including styrene–butadiene rubber, is largely dependent on the aging processes. Physical and chemical properties change, depending on the rubber content, the type of components used for the elastomer mixture and the resulting internal network structure. In the presented cases (SBR2 (10), SBR2 (20), SBR2 (30)) the decrease in tensile strength (TS) in relation to the samples not subjected to aging, and thus the observed increase in the relative elongation E_b_, may be caused by changes in the viscoelastic properties caused by the introduction of the filler in the form of ash BAC. The ash is characterized by a diverse mix of micro and macro elements that influence the oxidation processes of the composite structure. In addition to elevated temperature and humidity, oxygen, ozone, and light contribute to the aging processes. During the aging processes, the cross-links are degraded, causing the relaxation of the tight and more packed elastomeric structure, which, by changing the viscoelastic interactions, leads to an increase in the elongation parameters at the moment of breaking E_b_.

#### 4.2.6. Carbonyl Index

The Carbonyl index (CI), based on changes in the carbonyl (C=O) band, is used for to predict the lifespan and development of stabilizing additives [28,29,30,31]. The CI is used specifically to monitor the absorption band of carbonyl species formed during photooxidation or thermo-oxidation processes in the range from 1850 to 1650 cm^−1^ by measuring the ratio of the carbonyl peak to the reference peak. The carbonyl index is the ratio of the band height of the carbonyl groups to the band height of the C–H groups. An increase in CI indicates more advanced degradation. One of the most common analytical techniques for monitoring oxidation reactions, including CI, is Fourier transform infrared (FTIR) spectroscopy. FTIR analysis is also able to monitor other chemical changes that occur over the lifespan of the material, by detecting functional groups present in different bands.

The CI for the tested composites was calculated after temperature aging (Table 6). The addition of ash lowered the CI, so it can be concluded that chemical reactions took place in the vulcanizate structure, leading to the remodeling of individual bonds under the influence of temperature. Increasing the amount of filler in the matrix thus caused the matrix to degrade faster during processing. The lowest value (0.05) was recorded for the SBR1 sample (10).

#### 4.2.7. SEM Analysis

The morphology of the selected samples was assessed based on SEM photos taken at the breakthroughs of individual materials (Figure 6). Photographs at 250 times magnification revealed the diverse morphologies of the materials. In the case of the SBR reference sample, a small number of fine agglomerates can be observed, which may represent, e.g., the grains of the crosslinker. On the other hand, in the case of samples with the addition of ash, we can see an analogous increase in light agglomerates: the higher the addition of ash, the greater the proportion of particles with a regular structure, isolated from each other, with a wide size distribution.

#### 4.2.8. EDS Analysis

EDS analysis enabled qualitative and quantitative determination of the chemical compositions of the studied composites. However, it should be remembered that EDS provides only an approximate measure of individual minerals. The results are presented in Table 7. Aluminum and calcium were not observed in the SBR mix but appeared in the mixes with ash. This may indicate that these components were contained in the BCA1. The reference mixture also had a very low silicon content, whereas in SBR1 (10) and SBR1 (20) there was a significant increase in silicon content, which again leads us to the conclusion that silicon was a component of the ash.

#### 4.2.9. DSC & TGA Analysis

The results of thermogravimetric analysis (TGA) and differential scanning calorimetry (DSC) of selected SBR and SBR1 (10) composites are shown in Table 8, as well as in Figure 7 and Figure 8.

The addition of BCA filler did not significantly affect the temperature of thermal decomposition of SBR. As seen from the standard deviation, the maximum decomposition temperatures of all composites were the same. The total weight loss of the elastomers tested during degradation was 97% for the reference sample and 91% for the BCA-filled composites.

The glass transition temperature Tg of pure SBR is reported in the literature as 48/65 °C [32]. DSC showed that the glass transition temperature Tg starts at the beginning for the SBR composite, that is, at −46.23 °C, while for SBR1 (10) Tg starts at −46.92 °C (Figure 8). This suggests that the BCA filler acts as a plasticizer in SBR compounds. This effect is caused by certain amounts of macro- and microelements in the ashes obtained from brown coal combustion. These undoubtedly include calcium and magnesium, as well as silicon compounds. These elements, in combination with the components of the elastomer mixture, such as stearin or oxide activators, vulcanization accelerators, can impart plasticizing properties to rubber composites during cross-linking processes. The heat capacity of the SBR composite was 0.402 Jg^−1^K^−1^, whereas for SBR1 (10) the heat capacity decreased to 0.299 Jg^−1^K^−1^

#### 4.2.10. Testing Emission of VOCs from Vulcanizates

Ashes can adsorb different types of VOC’s, as well as gases or liquids in general. Ashes, depending on their physicochemical properties, can effectively adsorb various substances due to their high porosity, large surface area, suitable pore size and, for example, unburned carbon [33]. The larger surface area and carbon content with a large micropore volume results in higher adsorption capacity and longer penetration times. Additionally, the adsorption effect is related not only to the physicochemical properties of the adsorbent itself, but also to the physicochemical properties of the adsorbate, such as pore structure and surface functional groups [34].

The use of MCC-IMS allowed us to observe the effect of the ash on the release of chemical compounds from the mixture during the vulcanization process. The addition of a small amount of ash was found to reduce the amount of VOCs released during vulcanization, by approximately 48% on average (Table 9 and Figure 9). Ash as a highly porous and high-surface-area filler, can successfully trap released VOCs from rubber. The observed effect can be reduced by the polymer matrix which surround the filler particles “taking” their active groups on the surface, but also it can penetrate into the micro pores of the filler, reducing its number of free voids. With higher proportions of ash, decreases in emissions were also observed, but they were less significant. The reason for the higher VOCs emissions of the SBR1 sample (30) compared to the SBR1 sample (10) may be the amount of ash they contained.

Increasing the proportion of filler in the mixture causes agglomeration, which reduces the active surface area and, as a consequence, the adsorbing properties of the filler. In addition, agglomeration of the filler reduces its degree of dispersion, which may have a direct negative impact on the amount of VOCs released.

Two VOCs that were identified in this analysis are known as decomposition by-products of accelerators: benzothiazole and aniline. There was a reduction in the emission of two other unknown chemicals, based on comparison with the reference blend.

## 5. Conclusions

In this study, we used ash produced during the combustion of lignite at the Bełchatów Power Plant (Poland) in 2017 and 2018 as additives in SBR mixtures. The additives reduced the emission of volatile organic compounds (VOCs) from the rubber mixtures during vulcanization. Their use to fill elastomeric mixtures could reduce production costs, while also contributing to waste management of the ash.

The incorporation of ash into the structures of the prepared mixtures was confirmed by SEM, EDS, and FTIR spectroscopy. Differences in the morphology and chemical composition of the two analyzed ashes were also noted. These differences may result from the origin of the ash, the conditions of its formation, the combustion processes used and the place of sampling.

A decrease in the mechanical properties of the vulcanizates was observed following the addition of the ashes, which proves that they are inactive fillers. The addition of a small amount of BCA1 ash (10 pts. wt.) maintained the good mechanical properties of the SBR, while also reducing VOC emissions during vulcanization. The presented solution opens the way for doping with waste ash waste from the energy and mining industries, among others. The produced vucanizates can be used in the automotive industry, among others, for example in the production of rubber products used in closed spaces, e.g., car floor mats or other elements of car cabin equipment.

## Figures and Tables

**Figure 1 materials-14-04986-f001:**
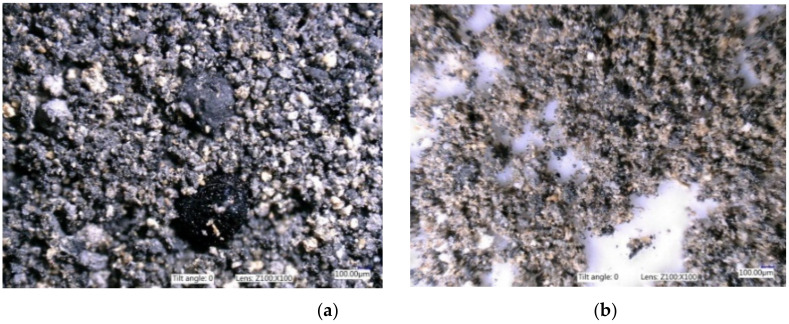
Images of BCA magnification ×1.00: (**a**) ash BCA1 and (**b**) ash BCA2.

**Figure 2 materials-14-04986-f002:**
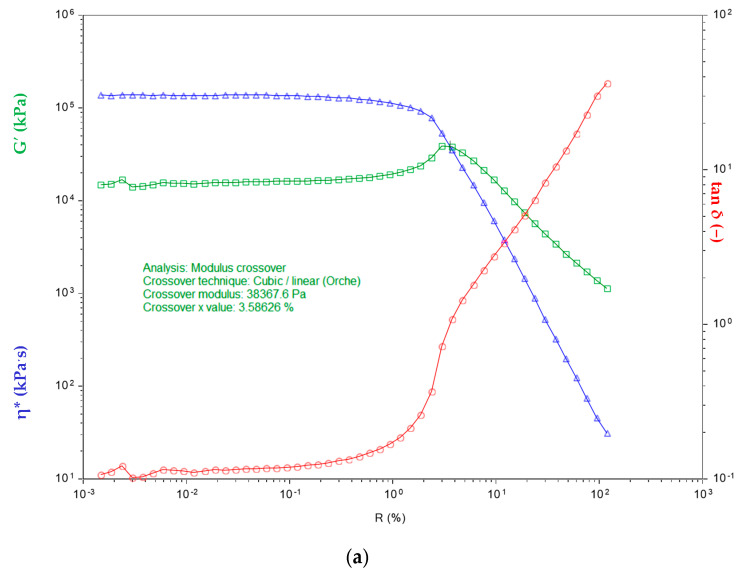
Dependence of viscosity and stress on the shear rate of the composites: (**a**) SBR; (**b**) SBR1 (10); (**c**) SBR2 (10); (**d**) SBR1 (20); (**e**) SBR2 (20). Sapphire curve —complex viscosity η* (kPa’s) at 80 °C (processing temperature) for SBR mixtures; green curve—storage shear modulus G′ (kPa) of uncured rubber mixtures SBR at 80 °C as a function of angular frequency ω (rad’s − 1) (linear viscoelastic region); red curve—mechanical loss tan δ (−) as a function of angular frequency ω (rad’s − 1) (oscillation strain 100%).

**Figure 3 materials-14-04986-f003:**
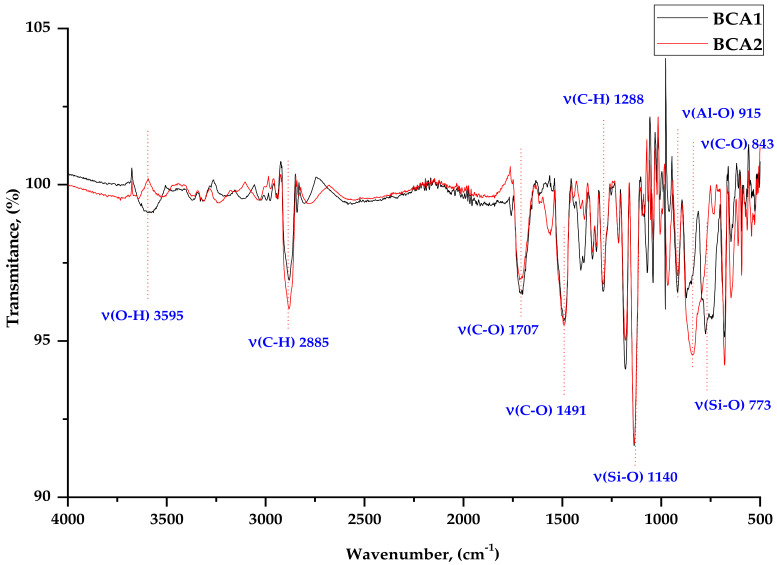
FTIR spectra for ashes BCA1 and BCA2.

**Figure 4 materials-14-04986-f004:**
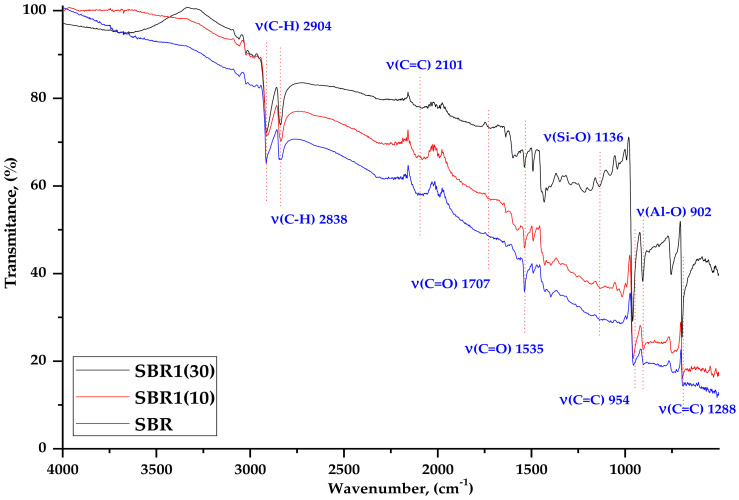
FTIR spectra of SBR, SBR1 (10), and SBR1 (30) composites.

**Figure 5 materials-14-04986-f005:**
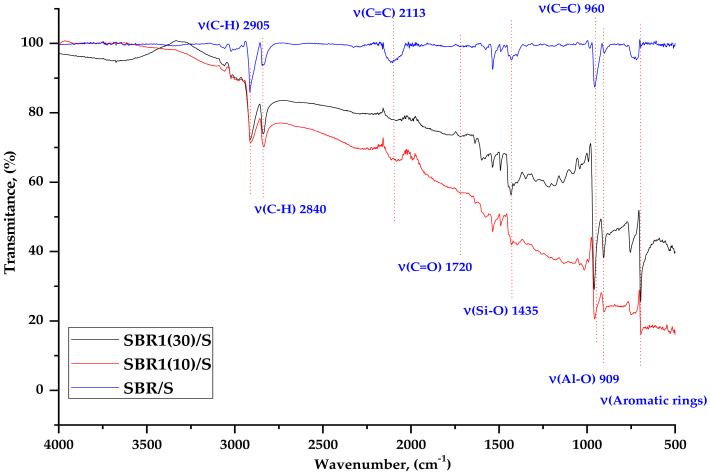
FTIR spectra for SBR, SBR1 (10), and SBR1 (30) after thermo-oxidative aging.

**Figure 6 materials-14-04986-f006:**
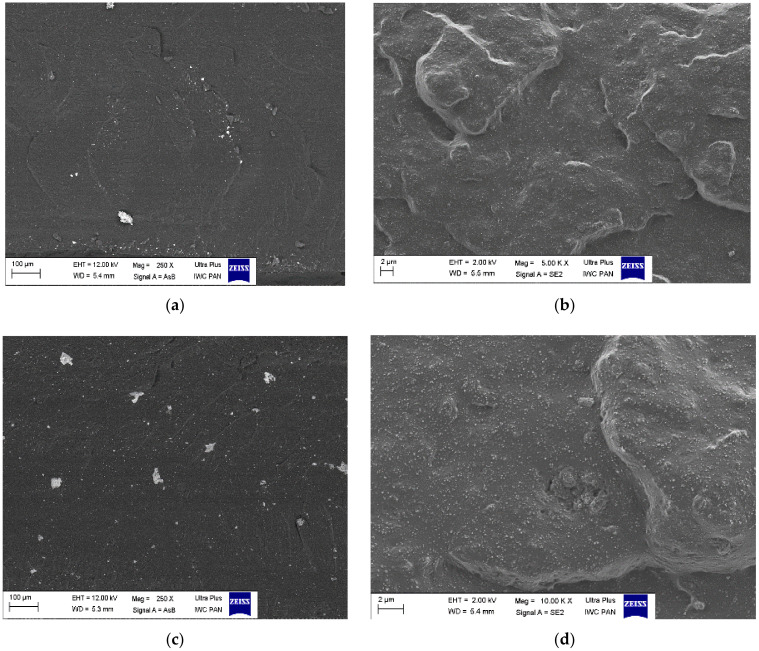
SEM photographs of composites (**a**,**b**)—SBR; (**c**,**d**)—SBR1 (10); (**e**,**f**)—SBR1 (20); left: photographs with ×250 magnification, right: photographs with ×5000 magnification.

**Figure 7 materials-14-04986-f007:**
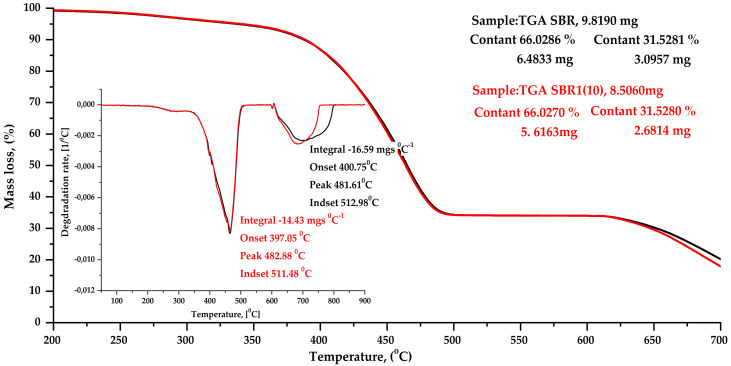
Thermogravimetric analysis (TGA) curves of SBR composites: reference sample SBR (black curve), SBR1 (10) sample (red curve).

**Figure 8 materials-14-04986-f008:**
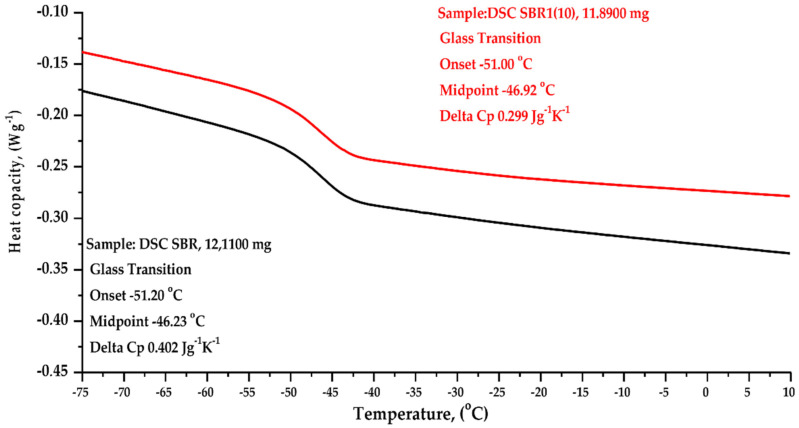
Comparative analysis of DSC results: unfilled SBR (black curve) and SBR10 (10 phr of BCA, red curve).

**Figure 9 materials-14-04986-f009:**
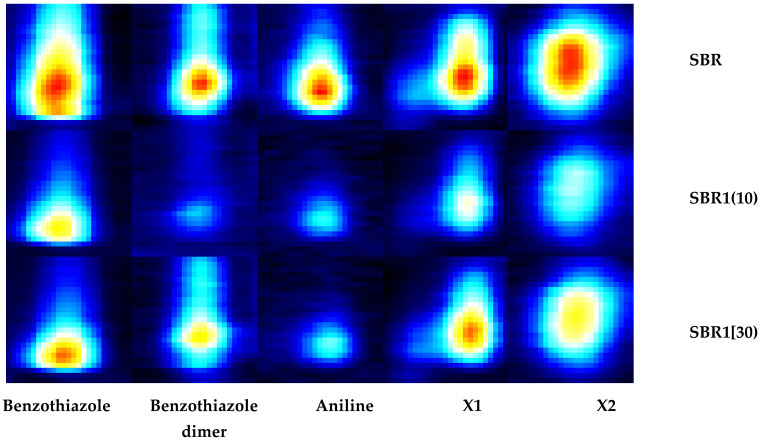
Graphical representation of VOCs released during the vulcanization process.

**Table 1 materials-14-04986-t001:** Compositions of the elastomer blends.

Symbol	SBR	SBR1 (10)	SBR1 (20)	SBR1 (30)	SBR2 (10)	SBR2 (20)	SBR2 (30)
SBR [phr]	100						
BCA1 [phr]	0	10	20	30	0	0	0
BCA2 [phr]	0	0	0	0	10	20	30
N330 [phr]	50	40	30	20	40	30	20
Stearin [phr]	1						
ZnO [phr]	3						
CBS [phr]	1						
Sulphur [phr]	2						

**Table 2 materials-14-04986-t002:** Ash sieve analysis BCA1 and BCA2.

	BCA1			BCA2		
Size Class [μm]	Δp [%]	q_3_ %[μm]	Δm [g]	Δp [%]	q_3_ %[μm]	Δm [g]
<45	8.1	0–18	4.30	23.6	0.52	11.79
45–63	9.2	0.51	4.59	26.1	1.45	13–05
63–125	38.2	0.62	119.00	27.2	0.44	13.63
125–250	36.5	0.29	18.18	17.3	0.14	8.68
250–500	5.8	0.02	2.91	4.3	0.02	2.15
500–1000	1.4	0.00	0.72	0.8	0.00	0.40
1000–2000	0.5	0.00	0.26	0.4	0.00	0.18
2000–4000	0.0	0.00	0.01	0.3	0.00	0.16
>4000	0.1	0.00	0.05	0.0	0.00	0.00

Legend: Δp—percentage distribution of the fraction share (%); q^3^%—density of the grain class distribution (μm); Δm—particle mass distribution (g).

**Table 3 materials-14-04986-t003:** Influence of BCA on the rheometric properties of SBR compounds.

Symbol	M_L_ [dNm]	∆M [dNm]	τ_90_ [min]
SBR	2.22	18.12	11.35
SBR1 (10)	1.90	17.11	9.67
SBR1 (20)	1.25	12.82	10.84
SBR1 (30)	1.37	12.68	12.99
SBR2 (10)	1.53	13.40	11.08
SBR2 (20)	1.52	13.03	11.36
SBR2 (30)	1.48	12.72	12.06

LL—minimum torque moment (dNm); ∆L—the decrease of torque moment (dNm) (∆L = LHR—LL); τ_02_—scorch time (min); τ_90_—time of vulcanization (min).

**Table 4 materials-14-04986-t004:** The cross-linking densities of the prepared mixtures.

Symbol	SBR	SBR1 (10)	SBR1 (20)	SBR1 (30)	SBR2 (10)	SBR2 (20)	SBR2 (30)
Total network density α [mol/cm^3^]	2.00 × 10^−4^	1.94 × 10^−4^	9.35 × 10^−5^	1.39 × 10^−4^	9.35 × 10^−5^	1.51 × 10^−5^	1.64 × 10^−4^

**Table 5 materials-14-04986-t005:** Influence of BCA filler on the mechanical properties of the SBR composites.

Symbol	SBR	SBR1 (10)	SBR1 (20)	SBR1 (30)	SBR2 (10)	SBR2 (20)	SBR2 (30)
H [ºSh]	60.85 ± 1.69	55.29 ± 2.63	55.39 ± 1.55	52.71 ± 1.08	54.43 ± 0.62	55.54 ± 1.12	51.83 ± 1.35
SE_100_ [MPa]	3.37 ± 0.20	2.63 ± 0.23	1.53 ± 0.06	1.49 ± 0.06	1.79 ± 0.03	1.86 ± 0.05	1.51 ± 0.06
SE_200_ [MPa]	8.47 ± 0.44	5.78 ± 0.52	2.44 ± 0.13	2.22 ± 0.13	3.48 ± 0.09	3.31 ± 0.17	2.21 ± 0.10
SE_300_ [MPa]	15.57 ± 0.75	10.66 ± 0.96	3.98 ± 0.30	3.39 ± 0.26	6.51 ± 0.18	5.98 ± 0.37	3.36 ± 0.18
T_S_ [MPa]	27.37 ± 0.32	18.5 ± 0.20	13.38 ± 0.66	11.86 ± 0.40	19.00 ± 0.35	15.95 ± 0.98	11.94 ± 0.54
E_b_ [%]	476 ± 24	441 ± 24	591 ± 25	619 ± 28	576 ± 15	535 ± 33	614 ± 14
Influence of theromoxidative and biological aging on the properties of SBR composites.							
Symbol	SBR	SBR1 (10)	SBR1 (20)	SBR1 (30)	SBR2 (10)	SBR2 (20)	SBR2 (30)
T_S_ [MPa]	23.16 ± 1.65	12.87 ± 1.46	10.28 ± 0.98	9.34 ± 0.54	16.98 ± 1.65	14.24 ± 0.40	6.89 ± 1.75
E_b_ [%]	282 ± 23	287 ± 18	370 ± 25	400 ± 16	423 ± 23	368 ± 17	422 ± 4

Legend: SE100 (MPa)—stress modulus with elongation at 100%; SE200 (MPa)—stress modulus with elongation at 200%; SE300 (MPa)—stress modulus with elongation at 300%; T_S_ (MPa)—tensile strength; E_b_ (%)—elongation at break.

**Table 6 materials-14-04986-t006:** Carbonyl index for samples after thermo-oxidative aging.

Symbol	Wavenumber [cm^−1^]		Carbonyl Index (CI)
	~1700	~2800	
SBR	0.92	6.30	0.15
SBR1 (10)	0.38	7.64	0.05
SBR1 (30)	0.91	8.80	0.10

**Table 7 materials-14-04986-t007:** Compositions of analyzed elastomer blends obtained by EDS analysis.

Chemical Elements, %	Designations of Mixtures		
	SBR	SBR1 (10)	SBR1 (20)
Oxygen	5.82	9.40	10.41
Sulfur	1.46	1.65	2.03
Zinc	1.27	0.58	1.74
Silicon	0.008	0.96	1.51
Aluminum	-	0.78	0.96
Calcium	-	1.54	1.59
Iron	-	-	1.28

**Table 8 materials-14-04986-t008:** Thermal characteristics of SBR and SBR1 (10).

Symbol	T_5%_ (°C)	T _peak 1 (DTG)_ (°C)	T _peak 2 (DTG)_ (°C)	Δm _total_ (%)	T_g_ (°C)
SBR	337	465	697	97.51	−46.23
SBR1 (10)	337	466.33	681	91.55	−46.92

T_5%_—decomposition temperature at 5% mass loss; T_p(DTG)_—temperature of maximum conversion rate on the DTG curve; Δm _total_—total mass loss during thermal decomposition; T_g_—glass transition temperature (standard deviations: T_5_, T_p (DTG_) ± 2 °C; Δm _total_ ± 0.6%; T_g_ ± 2 °C).

**Table 9 materials-14-04986-t009:** Comparison of the intensity of VOC signals released during the vulcanization process from the tested rubber mixtures.

	Benzothiazoles	Benzothiazole Dimer	Aniline	X1	X2
SBR	1129	399	532	973	466
SBR1(10)	961	171	178	608	264
SBR1(30)	1102	192	423	888	371

## Data Availability

Not applicable.

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
