# Peer review of "Use of Ashes from Lignite Combustion as Fillers in Rubber Mixtures to Reduce VOC Emissions"

_materials, 2021, doi:10.3390/ma14174986_

Round 1

Reviewer 1 Report

This paper illustrated “Use of ashes from lignite combustion as fillers in rubber mixtures to reduce VOC emissions”. The results are interesting, but many typing errors in this paper. It is a major revision needed. The comments are listed below. 1. Line 24, 86: Please pay attention to the subscript and superscript. So many typing errors. The authors should check the whole manuscript carefully. 2. Lines 24~33: Please check the order of reference! This is a serious typing error. 3. Except for "%" and "°C", there must be a space between the number and the unit. 4. Lines 169~170: “The heating time of the mixture at 160 °C was 2 × t90.” What is “2 × t90”? Some explanations are weird, please check the full text carefully. 5. Line 187: What is the magnification? Please represent by x…. 6. Lines 257, 269, 277: It’s wavenumber instead of wavelength. 7. Lines 292~293: The authors mentioned “The highest elongation was achieved by the SBR1 sample (30), with an increase of over 600% compared to the control (without filler).” Is 600% increasing in elongation break compared with the control (without filler)? It is an obvious editing errors, the authors should check again. 8. Page 14: (b) should be (d) in Figure 6! 9. Line 373: What is the abbreviation of BDC? 10. Line 374: It should be -48/65… 11. Lines 376~377: This suggests that BCA filler acts as a plasticizer in SBR compounds. How does the BCA plasticize the SBR? Please explain it. 12. Lines 378~379: What is Jgˆ1Kˆ1? Please check and correct it! 13. “Figure 10. Graphical representation of VOCs released during the vulcanization process.” Please rearrange Figure 10, the aniline, X1, and X2 are on wrong places. 14. Below Line 398, it should be Table 9. Please add it. 15. Line 381~395: How to reduce the amount of VOCs released during vulcanization by adding the ash? Please explain the mechanism clearly. 16. What is the composition of BCA? 17. The text in Figure 2 (a) ~ (d) is not clear. Please redraw it.

Author Response

Response

Reviewer 1

Thank you for making the review, we tried to comment in line with the guidelines. Thank you.

Q: 1. Line 24, 86: Please pay attention to the subscript and superscript. So many typing errors. The authors should check the whole manuscript carefully.

A: Throughout the text, superscripts and subscripts have been corrected.

Q: 2. Lines 24~33: Please check the order of reference! This is a serious typing error.

A: All literature in the text of the article has been corrected, also included in brackets in accordance with the guidelines in the journal template.

Q: 3. Except for "%" and "°C", there must be a space between the number and the unit.

A: Throughout the article, spaces between numeric values and units have been corrected.

Q: 4. Lines 169~170: “The heating time of the mixture at 160 °C was 2 × t90.” What is “2 × t90”? Some explanations are weird, please check the full text carefully.

A Value of t90 , so called optimum cure time – is a value measured by rheometer. For MCC-IMS analysis and during sample preparation, vulcanization process was occurred in closed reactor. Taking into consideration the fact that described way of vulcanization differ from conventional vulcanization process and to be sure that whole sample inside reactor is cured we increase time of vulcanization to two times optimum cure time (2 x t90).

Q: 5. Line 187: What is the magnification? Please represent by x….

A: Magnification corrected

Q: 6. Lines 257, 269, 277: It’s wavenumber instead of wavelength.

A:This  has been changed

Q: 7. Lines 292~293: The authors mentioned “The highest elongation was achieved by the SBR1 sample (30), with an increase of over 600% compared to the control (without filler).” Is 600% increasing in elongation break compared with the control (without filler)? It is an obvious editing errors, the authors should check again.

A: Elongation increases by 30% compared to the control sample - good point, corrected.

Q: 8. . Page 14: (b) should be (d) in Figure 6!

A: In Figure 6, the letters have been corrected.

Q: 9. Line 373: What is the abbreviation of BDC?

A: Incorrect abbreviation corrected.

Q: 10. Line 374: It should be -48/65…

A: This has been corrected.

Q: 11. . Lines 376~377: This suggests that BCA filler acts as a plasticizer in SBR compounds. How does the BCA plasticize the SBR? Please explain it.

A: This passage is explained in the text.

“This effect is caused by certain amounts of macro- and microelements in the ashes obtained from brown coal combustion. These undoubtedly include calcium and magnesium, as well as silicon compounds. These elements, in combination with the components of the elastomer mixture, such as stearin or oxide activators, vulcanization accelerators, can impart plasticizing properties to rubber composites during cross-linking processes.”

Q: 12. Lines 378~379: What is Jgˆ1Kˆ1? Please check and correct it!

A: This has been corrected.

Q: 13. “Figure 10. Graphical representation of VOCs released during the vulcanization process.” Please rearrange Figure 10, the aniline, X1, and X2 are on wrong places.

A: Figure 10 has been corrected.

Q: 14. Below Line 398, it should be Table 9. Please add it.

A: This has been added. Q: 15. Line 381~395: How to reduce the amount of VOCs released during vulcanization by adding the ash? Please explain the mechanism clearly.

A: Explanations in the text have been added. Ashes are well known materials (particles) which can adsorb different type of VOC’s but also gases or liquids. Ash, depends on their physicochemical properties, can effectively adsorb different substances due to its high porosity, high surface area, appropriate pore size and e.g., unburned carbon [1]. Higher surface area and carbon content having high micropore volume results in a higher adsorption capacity and larger breakthrough time. In addition, the adsorption effect is not only related to the physicochemical properties of the adsorbent itself but also to the physicochemical properties of the adsorbate, such as pore structure, surface functional groups [2]. Taking into consideration that ash used as a filler in our study is surrounded by polymer matrix, can reduce adsorbing property of ash, however presented results showed successfully reduce the amount of VOCs released from rubber.

Q: 16. What is the composition of BCA?

A: BCA - Brown Coal Ash. The BCA abbreviation was introduced in the abstract

Q: 17. The text in Figure 2 (a) ~ (d) is not clear. Please redraw it.

A: The graphical curves in Figure 2 have been corrected.

Reviewer 2 Report

In this related article, the authors used ashes from lignite combustion as fillers in rubber mixtures to reduce VOC emissions. Seven different mixtures were produced by a roller mixing mill. Properties of mixtures were characterized by mechanical, rheological test and biological aging. VOCs were studied using ion mobility spectroscopy with gas chromatography.

My comments, suggestions, and questions are the following:

  • In row 187. The designation of pictures is the same (see (a) ash BCA1 (b) ash BCA1.).
  • In Fig. 2. the axes designation is not correct.
  • In Fig. 4. can not be any trends. How could the Authors explain it?
  • In Fig3. appear BDC abbreviation? What does it mean?
  • In row 239. in the text is Fig 4. without any figure.
  • In Fig 6. appear two b) designation.
  • In Table 5. How can the Authors explain the trend change between Eb[%] SBR2(10) and SBR2(20)? After at the influence of theromoxidative and biological aging the trend of properties (Eb[%]) of SBR composites was changed between SBR2(20) and SBR2(30) too, what is the reason?
  • In row 341. the designation of b), d), f) was not explained.
  • Why did only two material’s results appeared in Fig 7. and Fig 8.?
  • Why did only two material’s results appeared in Fig 9. and Fig 10.?
  • The font size in the text is changing hectically.

Author Response

Response

Reviewer 2

Thank you for making the review, we tried to comment in line with the guidelines. Thank you.

Q: 1. In row 187. The designation of pictures is the same (see (a) ash BCA1 (b) ash BCA1.).

A: This has been corrected.

Q: 2. In Fig. 2. the axes designation is not correct.

A: This has been corrected.

Q: 3. In Fig. 4. can not be any trends. How could the Authors explain it?

A: Description has been changed.

The intensity of the BCA bands changes with the amount of ash, although these changes are not as significant. Small changes are visible in the dactyloscopic area related to a slight increase in the intensity of the bands in the range of 1500-1400 cm-1 and 500 cm-1. The first are characterized by stretching vibrations of the −C = C− aliphatic grouping and bending vibrations = C − H. However, the second is also the confirmation of the occurrence of stretching vibrations of the aromatic ring v (C = C), which come from styrene.

Q: 4. In Fig3. appear BDC abbreviation? What does it mean?

A: This has been corrected.

Q: 5. In row 239. in the text is Fig 4. without any figure.

A: This has been corrected.

Q:6. In Fig 6. appear two b) designation.

A: : This has been corrected.

Q: 7. In Table 5. How can the Authors explain the trend change between Eb[%] SBR2(10) and SBR2(20)? After at the influence of theromoxidative and biological aging the trend of properties (Eb[%]) of SBR composites was changed between SBR2(20) and SBR2(30) too, what is the reason?

A: Clarifications have been made in the text for the segment on aging of the

The lifetime of elastomers, including styrene-butadiene rubber, is largely dependent on the aging processes. Physical and chemical properties change, depending on the rubber content, the type of components used for the elastomer mixture and the resulting internal network structure. In the presented cases (SBR2 (10), SBR2 (20), SBR2 (30)) the decrease in tensile strength TS in relation to the samples not subjected to aging, and thus the observed increase in the relative elongation Eb, may be caused by changes in the viscoelastic properties caused by the introduction of the filler in the form of ash BAC. The ash is characterized by a diverse content of micro and macro elements that influence the oxidation processes of the composite structure. In addition to elevated temperature and humidity, oxygen, ozone and light contribute to the aging processes. During the aging processes, the cross-links are degraded, causing the relaxation of the tight and more packed elastomeric structure, which, by changing the viscoelastic interactions, leads to an increase in the elongation parameters at the moment of breaking Eb.

Q: 8. In row 341. the designation of b), d), f) was not explained.

A: This has been clarified.

Q: 9. Why did only two material’s results appeared in Fig 7. and Fig 8.?

A: Changes in phase transitions and in weight loss were determined only for selected composites containing ashes, which showed more favorable physical and chemical parameters.

Q: 10. Why did only two material’s results appeared in Fig 9. and Fig 10.?

A: Due to the use of advanced analysis of ion mobility spectroscopy with gas chromatography, we focused on selecting only some composites for this study.

Q: 11. The font size in the text is changing hectically.

A: The part size throughout the text has been corrected in line with the Materials MDPI template guidelines.

Round 2

Reviewer 1 Report

Line 178: Why is there subscript 1 in DSC?

Line 262: Page 12 is blank, please check it.

If the authors finish the above minor comment, this paper is ready for publication.

Author Response

Line 178: 1 w nazwie DSC1, to nazwa aparatu.

Line 262: Usunięta pusta strona.
